# Regulation of Interferon Induction by the Ubiquitin-Like Modifier FAT10

**DOI:** 10.3390/biom10060951

**Published:** 2020-06-23

**Authors:** Mei Min Mah, Nicola Roverato, Marcus Groettrup

**Affiliations:** 1Division of Immunology, Department of Biology, University of Konstanz, 78457 Konstanz, Germany; Mei.Mah@uni-konstanz.de (M.M.M.); Nicola.Roverato@uni-konstanz.de (N.R.); 2Biotechnology Institute Thurgau at the University of Konstanz, 8082 Kreuzlingen, Switzerland

**Keywords:** interferon, ubiquitin, FAT10, signal transduction, RIG-I, influenza virus

## Abstract

The revelation that the human major histocompatibility complex (MHC) class I locus encodes a ubiquitin-like protein designated HLA-F adjacent transcript 10 (FAT10) or ubiquitin D (UBD) has attracted increasing attention to the function of this protein. Interestingly, the pro-inflammatory cytokines interferon (IFN)-γ and tumor necrosis factor (TNF) α synergize to strongly induce FAT10 expression, thereby suggesting a role of FAT10 in the immune response. Recent reports that FAT10 downregulates type I interferon production while it upregulates IFN-γ pose mechanistic questions on how FAT10 differentially regulates interferon induction. Several covalent and non-covalent binding partners of FAT10 involved in signal transduction pathways leading to IFN synthesis have been identified. After introducing FAT10, we review here recent insights into how FAT10 affects proteins in the interferon pathways, like the virus-responsive pattern recognition receptor RIG-I, the ubiquitin ligase ZNF598, and the deubiquitylating enzyme OTUB1. Moreover, we outline the consequences of FAT10 deficiency on interferon synthesis and viral expansion in mice and human cells. We discuss the need for covalent isopeptide linkage of FAT10 to the involved target proteins and the concomitant targeting for proteasomal degradation. After years of investigating the elusive biological functions of this fascinating ubiquitin-like modifier, we review the emerging evidence for a novel role of FAT10 in interferon regulation.

## 1. Introduction

HLA-F adjacent transcript 10 (FAT10) is a ubiquitin-like modifier (ULM) that is encoded in the major histocompatibility complex (MHC) [1] and is induced in mammals in virtually all tissues following synergistic stimulation with the inflammatory cytokines tumor necrosis factor α (TNFα) and interferon γ (IFN-γ) [2,3,4]. Recent efforts to solve the structure of FAT10 by NMR and X-ray crystallography confirmed that the two domains of FAT10 are arranged in a tandem head-to-tail formation joined by a short flexible linker [5,6]. In contrast to ubiquitin that requires posttranslational cleavage to expose its terminal diglycine motif, FAT10 is synthesized with a readily accessible diglycine motif at its C-terminus [7]. Moreover, while ubiquitin is expressed ubiquitously, the cellular abundance of FAT10 is tightly controlled by both transcriptional and posttranslational regulation. This confines the basal expression of FAT10 mainly to organs of the immune system like thymus, lymph nodes, and spleen [8,9,10]. In the growing list of ULMs, FAT10 is the only member that directly targets proteins for proteasomal degradation independently of ubiquitin attachment [11,12,13]. FAT10 gets conjugated via isopeptide linkage to its target proteins through the E1 enzyme UBA6 (UBE1L2, MOP4) and the E2 enzyme USE1 (UBE2Z) [14,15,16], while neither E3 ligases nor FAT10 deconjugating enzymes have been reported so far. FAT10 docks to the 26S proteasome via its subunit RPN10 (S5a) and leads the bulk of its conjugates to proteasomal degradation with an overall half-life of one hour [17]. In contrast to ubiquitin, FAT10 does not get cleaved from its substrates at the 26S proteasome but is degraded along with its substrates, which explains its short half-life. Proteomic analyses performed by Aichem et al. [18] revealed that endogenous FAT10 interacts with more than 500 proteins involved in diverse functional pathways and is attached via isopeptide linkage to about 150 of them. Indeed, FAT10 has been implicated in various cellular functions like apoptosis [19,20], the spindle check point during the mitotic cell cycle [2,21,22], and NF-κB activation [23]. Furthermore, FAT10 is highly expressed in a multitude of different human tumors, and the use of mass spectrometry analysis and co-immunoprecipitation (IP) studies showed that FAT10 interacts with a broad spectrum of proteins that are involved in tumorigenesis (reviewed in [24]). 

Gradually, works undertaken by researchers around the world have started to shed light on the biological functions of FAT10. The generation of FAT10 knockout mice revealed that the mice are viable and fertile, indicating that FAT10 deficiency does not interfere with essential housekeeping tasks [8]. However, mice lacking FAT10 were reported to be hypersensitive towards lipopolysaccharides, while their lymphocytes were more susceptible to spontaneous apoptotic cell death. Subsequent investigations of older FAT10^-/-^ mice revealed that FAT10 deficiency increased the life span of mice by 20%, enhanced their metabolic rate, mainly fat consumption, and led to a marked reduction of adipose tissue [25]. Interestingly, the expression of the anti-inflammatory cytokine IL-10 was significantly enhanced in the muscle and plasma of FAT10^-/-^ mice, which may be causally linked to the reduction of white fat tissue, as it is known to be propagated by inflammation. 

Strikingly, FAT10 is by far most prominently expressed in the epithelial cells of the thymic medulla in mice and humans, where it modulates the thymic negative selection of T cells [26]. In this context, FAT10 expression has been observed to influence the repertoire of the eluted MHC class I peptide ligands, giving a first hint at its function in immunity. Furthermore, in 2014, Spinnenhirn et al. observed that in the presence of intracellular bacteria such as Salmonella, FAT10 co-localizes with the autophagy adapter p62 to decorate the pathogen in the early phase of intracellular xenophagy, suggesting that FAT10 may be directly involved in antibacterial host defense [27]. Additionally, several recent works have provided evidence that FAT10 is involved in the antiviral immune response. In this article, we review the role of FAT10 in the antiviral immune response in vivo and in vitro. In particular, we will discuss the current understanding of how FAT10 regulates the RIG-I signaling pathway, which results in altered interferon secretion and an impaired response to viral infections.

## 2. FAT10 Modulates IFN Production

A regulated protein turnover by the ubiquitin proteasome system is important for immune regulation of eukaryotic cells, as it governs the majority of cellular processes including antigen processing, immune response, and inflammation. Over the years, members of the ULM family such as SUMO and ISG15 have been shown to be modulators of the innate and adaptive immune responses (reviewed in [28]). Likewise, FAT10 was discovered in 1996 in conjunction with an effort to identify additional genes from the human MHC, which hinted at its involvement in immunity [2]. Moreover, its cytokine inducibility [3,4,5], immune-related expression profile [8,29,30], and upregulation in mature dendritic cells [7,30] and macrophages [31] point towards the direction that FAT10 plays a role in the immune system. This is further supported by the findings that the combined stimulation with TNFα and IFN-γ leads to FAT10 induction in cells of numerous tissues [4]. For example, infection of C57BL/6 mice with lymphocytic choriomeningitis virus (LCMV) known to systemically induce TNFα and IFN-γ by NK cells, cytotoxic T lymphocytes, and Th1 cells leads to prominent FAT10 mRNA induction [10]. Splenocytes expressed higher levels of FAT10 already at day 3 post-LCMV infection, whereas the thymus showed higher amounts of FAT10 mRNA only by day 8 after LCMV infection. By assessing the cytokine profile of FAT10^-/-^ mice subsequent to LCMV infection, it was observed that FAT10^-/-^ mice secreted more IFN-α and IFN-β in comparison to FAT10^+/-^ mice. On the contrary, FAT10^-/-^ mice produced lower levels of IFN-γ after T cell receptor (TCR) stimulation (Figure 1) [10]. This finding suggested that FAT10 serves to fine-tune the balance between type I and type II IFN production during viral challenge. Besides, these data provided evidence for a role of FAT10 in antiviral defense in an in vivo setting. 

An involvement of FAT10 in the innate antiviral defense to virus infection was also found by Zhang and colleagues when studying the replication of influenza A virus (IAV) and type I interferon production in the human lung epithelial cell line A549. To begin with, cellular mRNAs and proteins that were up- or downregulated in response to influenza A virus infection were determined [32]. Remarkably, among 15 ubiquitin-like proteins, the mRNA for FAT10 was up-regulated most vigorously when A549 cells were infected with an H5N1 strain of avian IAV. *Fat10* mRNA expression peaked at 24 h post-H5N1 infection and decreased gradually by 48 and 72 h post-infection, both in mouse lungs and in the human lung adenocarcinoma cell line A549. The high FAT10 expression upon viral infection was correlated with the promotion of virus-induced cell death, as seen in A549 and HBEpic cells. A role for FAT10 in promoting IAV expansion was confirmed by a series of FAT10 knockdown experiments. In the absence of FAT10, A549 cells showed less IAV-induced cell death and a lower virus load. When searching for the mechanism by which the ablation of FAT10 suppressed IAV replication, Zhang et al. found that the knock down of *Fat10* significantly enhanced the production of the type I interferons IFN-α and IFN-β by IAV-infected cells [32]. In a nutshell, the high *Fat10* expression induced through viral RNA during H5N1 infection led to blunted IFNα/β release and enabled an enhanced viral replication. This observation is in agreement with increased type I IFN secretion from the splenocytes of LCMV-infected FAT10^-/-^ mice as compared to wild-type mice [10], thus strengthening the hypothesis that FAT10 mitigates virus-induced IFNα/β generation, possibly to limit inflammation and associated tissue damage.

Nevertheless, while there was a significant effect of FAT10 induction on IAV replication in A549 cells in vitro, FAT10^-/-^ mice infected intranasally with IAV showed no differences in IFN-α, IFN-β, and IFN-γ secretion in splenocytes and lung cells as compared to their heterozygous littermates. The only observable difference between IAV-infected FAT10-proficient and -deficient mice was that FAT10^-/-^ mice displayed a 50% enhanced IFN-α concentration in the serum on day 4 after IAV infection. Nevertheless, the IAV titer in the lung, the loss of body weight, and the survival after a semi-lethal IAV infection did not differ between FAT10^-/-^ and FAT10^+/-^ mice [10]. Apparently, the FAT10-mediated in vitro effects observed in A549 cells may not be strong enough to exert differences in disease symptoms after influenza infection in mice. Likewise, FAT10^-/-^ mice eliminated the strain LCMV-WE after experimental infection, with a normal kinetic and decline in virus titer that did not differ from those observed for wild-type mice [10]. In contrast, when NRAMP1 transgenic C57BL/6 mice were orally infected with *Salmonella typhimurium*, an increase in the bacterial burden in the mesenterical lymph nodes of FAT10-deficient mice was observed on day 14 after infection as compared to FAT10-proficient control mice [27]. In addition, FAT10^-/-^ mice showed a slightly reduced survival rate and an increased loss of body weight after *Salmonella* infection. Since IFN-γ is pivotal for the immune response to *Salmonella* bacteria [33], it would be interesting to investigate if a reduced IFN-γ response in the face of FAT10 deficiency, as found for splenocytes from LCMV-WE-infected mice, contributed to the enhanced susceptibility of FAT10^-/-^ mice to infection with *S. typhimurium*. 

## 3. FAT10 Impairs the Type I Interferon Signaling Cascade

The observed FAT10-dependent reduction of IFNα/β production by H5N1 influenza virus-infected A549 cells poses the question of how FAT10 might exert this effect mechanistically. In principle, FAT10 could interfere with the intracellular recognition of the virus or with the signal transduction pathway emanating from the involved pattern recognition receptors. There are two distinct signaling pathways involved in type I IFN production, which are distinguished by the type of viral sensors, namely, the Toll-like receptors (TLRs) and the RIG-I-like receptors (RLRs) (Figure 2). In the first pathway, the viral double-stranded (ds)RNA is recognized in the endosome by TLR3. Here, the activated TLR3 recruits the E3 ligase TNF receptor-associated factor (TRAF)3 which in turn poly-ubiquitylates itself leading to its functional activation (Figure 2). Furthermore, in the cytosol, the viral dsRNA is detected by the RNA sensor RIG-I, which consequently undergoes conformational changes that trigger the TRIM25-mediated ubiquitylation of RIG-I and the subsequent recruitment and activation of TRAF3 (discussed in Section 6) [34,35,36]. Both receptors function as sensors that recognize foreign RNA in a distinct manner, which eventually leads to the activation of either the NF-κB pathway or the TBK1/IKKε complex axis, resulting in secretion of pro-inflammatory cytokines or type I IFN, respectively (Figure 3) [37]. Structurally, FAT10 bears a diglycine motif at its C-terminus that facilitates the isopeptide linkage of FAT10 to hundreds of different substrates, which is termed FAT10ylation [18,19]. FAT10ylation is mediated by an E1, E2, and possibly an E3 enzymatic cascade, where UBA6 [14,16] and USE1 [38] serve as E1-type activating and E2-type conjugating enzymes, respectively, and so far unknown E3 ligases may confer specificity in substrate selection. Zhang and colleagues showed that the knockdown of the E1 enzyme of FAT10, UBA6, or the FAT10 E2 enzyme, USE1, resulted in reduced mRNA levels of the IAV-encoded M1 protein as compared to wild-type A549 cells, suggesting that covalent FAT10ylation of viral or cellular proteins is required to promote H5N1 viral replication [32]. Further investigations into the viral components of the H5N1 virus showed that the viral RNA pool was responsible for inducing high FAT10 expression. The viral RNA was sensed by the cytoplasmic RNA sensor RIG-I, which then activated the downstream transcription factor NF-κB that switched on *Fat10* gene expression (Figure 3). The proposed mechanism was supported by data showing that FAT10 expression was repressed by knockdown of RIG-I and p65 in A549 cells. Conversely, the overexpression of RIG-I and p65 resulted in enhanced FAT10 promoter-driven luciferase activity. Therefore, upregulated FAT10 facilitated viral replication which served as a positive feedback loop in the RIG-I–NF-κB–FAT10 signaling pathway. Since IFNα/β bind to their common receptor IFNAR1, which subsequently induces phosphorylation of STAT1 and STAT2, the knockdown of FAT10 consistently enhanced STAT1 phosphorylation [32]. 

## 4. Mechanistic Insights into how FAT10 Attenuates RIG-I-Mediated Signal Transduction

An obvious starting point by which FAT10 might impair RIG-I-mediated signal transduction leading to IFNα/β induction is non-covalent binding or covalent conjugation to RIG-I itself. While studying the formation of cytoplasmic RIG-I-containing stress granules upon infection of cells with Sendai virus or stimulation with the RIG-I ligand poly(I:C), Joo-Yeon Yoo and colleagues found that the joint stimulation of infected cells with TNFα and IFN-γ interfered with RIG-I granule formation and that FAT10 was the cytokine-induced factor that caused this effect [39]. Overexpression of FAT10 in cells that were stimulated with the RIG-I ligands poly(I:C) or 5’ppp-dsRNA resulted in reduced RIG-I granule formation as well as reduced RIG-I-dependent activation and nuclear translocation of the transcription factors interferon regulatory factor-3 (IRF-3) and NF-kB. These effects of wild-type FAT10 were also observed when a conjugation-incompetent variant of FAT10 was used, in which the C-terminal GG motif was replaced by AA, suggesting that non-covalent binding by FAT10 rather than FAT10 conjugation was underlying these effects. As endogenous FAT10 and RIG-I co-localized in TNFα/IFN-γ-stimulated cells, a direct interaction of FAT10 and RIG-I was investigated [39]. Indeed, FAT10 was shown to non-covalently interact with the two caspase activation and recruitment domains (CARD) of RIG-I. In spite of FAT10 being a protein-targeting signal, endogenous or ectopic FAT10 expression did not target RIG-I for degradation but caused RIG-I to become insoluble, leading to reduced RIG-I co-localization with the mitochondrial antiviral signaling (MAVS) protein at the surface of mitochondria (Figure 4). As the assembly of RIG-I with MAVS at mitochondria is pivotal for the down-stream activation of NF-kB and IRF3, this could be a mechanism by which FAT10 downregulates type I interferon induction, which relies on IRF3 phosphorylation and nuclear translocation. 

## 5. TRIM25 and ZNF598—Ubiquitin Ligases Involved in the Regulation of RIG-I Signaling by FAT10

Another remarkable finding by the Yoo group was that the E3 ligase TRIM25, which binds and poly-ubiquitylates RIG-I after activation, also binds to FAT10 and stabilizes it by reducing its degradation rate [39]. After binding of viral dsRNA to the C-terminal helicase domain of RIG-I, its N-terminal CARD domains are exposed and subsequently covalently modified with K63-linked poly-ubiquitin chains by the RING-type ligase TRIM25 [40,41]. RIG-I poly-ubiquitylation promotes the subsequent association of RIG-I with MAVS and downstream signaling to induce type I interferon production. In principle, the binding of FAT10 to TRIM25 and to RIG-I via its CARD domains could lead to reduced RIG-I signaling. Nevertheless, whether FAT10 expression affects the poly-ubiquitylation of activated RIG-I by TRIM25 was not addressed in this study. Therefore, it remains to be investigated which role TRIM25 inactivation or the FAT10-mediated sequestration of RIG-I into not yet described insoluble compartments might have under endogenous conditions in RIG-I signaling (Figure 4). In contrast to the tightly folded and readily soluble ubiquitin, FAT10 easily unfolds and precipitates in cells, at least under overexpression conditions and as a recombinant protein in vitro, and these FAT10 aggregates are not easily dissolved [5,42,43]. Therefore, overexpression studies with FAT10 have to be carefully controlled and confirmed under endogenous conditions in order to avoid artefacts that might result from the unphysiological precipitation of FAT10 [44]. A priori, it seems difficult to imagine how co-precipitation of RIG-I and FAT10 into an insoluble cellular compartment can be a means of regulating signal transduction in a reversible manner, given that reversibility conveyed by highly regulated pairs of cognate enzymes like kinases/phosphatases or ubiquitin ligases/de-ubiquitylating enzymes is a hallmark of signaling pathways. Nevertheless, sequestration, especially of intrinsically disordered proteins, into storage granules and liquid phase separation in cells are an emerging topic. Whether and how FAT10 might use such mechanisms in order to regulate signal transduction requires further investigations.

The notion that complex formation between RIG-I and FAT10 affects type I interferon signaling has recently been corroborated by a study investigating another RING-type ubiquitin E3 ligase named ZNF598 [45]. Oshiumi and colleagues found that ZNF598, which was known to dampen the induction of type I interferon-stimulated genes upon viral infection [46], attenuated the induction of type I interferons upon ZNF598 overexpression. Conversely, silencing the *ZNF598* gene augmented IFN-β expression upon poly(I:C) stimulation of cells. As overexpression of ZNF598 unexpectedly reduced the poly-ubiquitylation of activated RIG-I, it was investigated if the non-covalent binding of FAT10 to RIG-I might be involved in this regulation. Indeed, it was found that ZNF598 binds to both FAT10 and RIG-I and that the FAT10–RIG-I complex was stabilized by association with ZNF598, while ZNF598 silencing weakened the binding of FAT10 to RIG-I in TNFα/IFN-γ-stimulated cells (Figure 4). The upregulation of IFN-β signaling after ZNF598 silencing did not occur in FAT10-deficient cells, indicating that ZNF598 exerted its interferon-dampening effects via FAT10. Interestingly, the poly-ubiquitylation of RIG-I upon poly(I:C) stimulation was shown to be strongly increased in FAT10-deficient cells, whereas evidence that FAT10 expression affected the steady-state level of soluble RIG-I was not obtained in this study [45]. Hence, it appears more likely that under physiological conditions, FAT10 suppresses IFNα/β signaling by binding to the CARD domains of RIG-I and interfering with RIG-I ubiquitylation, rather than by rendering it insoluble.

## 6. Further Down the Interferon Induction Cascade: FAT10 Activates the Deubiquitylating Enzyme OTUB1

The formation of RIG-I/MAVS multimers at the surface of mitochondria leads to binding and activation of the E3 ligase TRAF3, which auto-modifies itself with K63-linked poly-ubiquitin chains. Poly-ubiquitylated TRAF3 then binds and activates the kinases TANK-binding kinase (TBK)1 and inhibitor of nuclear factor κ-B kinase (IKK)ε. These kinases phosphorylate the transcription factors IRF3 and IRF7, which then translocate into the nucleus to induce IFN-β and IFN-α transcription (Figure 3). TRAF3 poly-ubiquitylation is downregulated by a deubiquitylating enzyme otubain 1 (OTUB1) which belongs to the ovarian tumor (OTU) family of cysteine proteases [47]. OTUB1 downregulates ubiquitylation not only enzymatically by cleavage of ubiquitin chains but also by non-covalently binding to the ubiquitin-conjugating enzymes UbcH5b and Ubc13, which leads to their inhibition [48]. During an interaction screen for proteins binding to USE1, i.e., the E2 enzyme of FAT10, OTUB1 was identified and validated as a direct binding partner of USE1. Moreover, OTUB1 was found to bind non-covalently and covalently to FAT10 as well (Figure 5). A small portion of OTUB1 is FAT10ylated in cells, which leads to its degradation by the 26S proteasome [49]. Unexpectedly, the prominent non-covalent binding of FAT10 to OTUB1 leads to both the activation of the deubiquitylating activity of OTUB1 and the non-enzymatic inhibition of E2 enzymes. In particular, it was shown that FAT10 binding to either wild-type OTUB1 or the enzymatically inactive C91S variant of OTUB1 downregulated the poly-ubiquitylation of TRAF3 in cells, and this effect was especially prominent for ubiquitin chains linked via the lysine residue 63 of ubiquitin (K63) [49]. Although the contribution of the FAT10-mediated activation of OTUB1 to the induction of IFNα/β remains to be investigated, it is likely that the observed reduced TRAF3 ubiquitylation in the presence of FAT10 will limit type I interferon production by virus-infected cells and consequently attenuate ensuing inflammation and tissue damage.

## 7. Concluding Remarks and Perspectives

While progress in understanding the covalent conjugation and function of FAT10 as a signal for degradation by the 26S proteasome has been substantial in the past years, the biological function of FAT10 is still poorly understood. Similar to many genes of the immune system, the inactivation of the *Fat10* gene in mice produces only mild phenotypes. Three groups have independently shown in mouse and human cell lines and primary cells that FAT10 significantly downregulates type-I interferon induction after infection with Sendai, influenza, and LCMV viruses (Figure 6). At least in human lung epithelial cells, FAT10 deficiency led to increased IFNα/β production, which caused a more efficient suppression of influenza virus replication. However, the experimental intranasal infection of mice with a semi-lethal dose of a highly pathogenic strain of influenza A virus did not show any differences in the virus titers or in disease symptoms between wild-type and FAT10-deficient mice. Following the hypothesis that FAT10 limits type I interferon production in order to avoid overshooting inflammation and tissue destruction, it will be pertinent to investigate if FAT10^-/-^ mice show enhanced tissue damage and inflammatory infiltrates in the lung after infection with influenza A virus also at lower doses. Moreover, it will be important to infect FAT10^-/-^ mice with further RNA viruses to determine whether differences in the course and severity of infectious diseases can be recorded as compared to wild-type mice.

In mechanistic terms, there is agreement that the dsRNA sensor RIG-I, that is central for the type I interferon response to influenza virus infection, is non-covalently bound by FAT10 and that this interaction is functionally relevant for the suppression of IFNα/β induction by FAT10. Whether FAT10 mediates this effect by sequestering RIG-I into insoluble aggregates or a distinct liquid phase or by interfering with the poly-ubiquitylation of RIG-I remains to be further investigated (Figure 6). Also the contributions of FAT10-mediated activation of OTUB1 and the increased de-ubiquitylation of TRAF3 to a diminished transcriptional induction of the genes for IFNα/β will need to be investigated. After all, at least one functional consequence of the massive induction of FAT10 by TNFα and IFN-γ in infected tissues has emerged that deserves further elucidation by research efforts in the years to come.

## Figures and Tables

**Figure 1 biomolecules-10-00951-f001:**
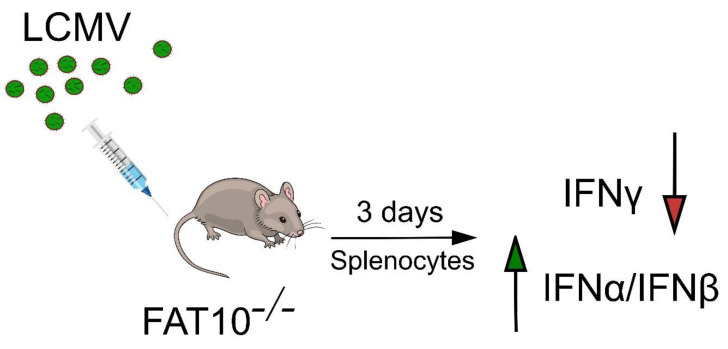
HLA-F adjacent transcript 10 (FAT10) balances the interferon response in vivo. FAT10-deficient mice display an altered interferon response 3 days after infection with lymphocytic choriomeningitis virus (LCMV). Splenocytes from LCMV-infected FAT10^-/-^ mice show a reduced production of IFN-γ and an enhanced production of IFNα/β, indicating that FAT10 expression induced by viral infection enhances the type II interferon response while it attenuates the type I interferon response. The latter effect may limit inflammation and tissue damage in infected tissues.

**Figure 2 biomolecules-10-00951-f002:**
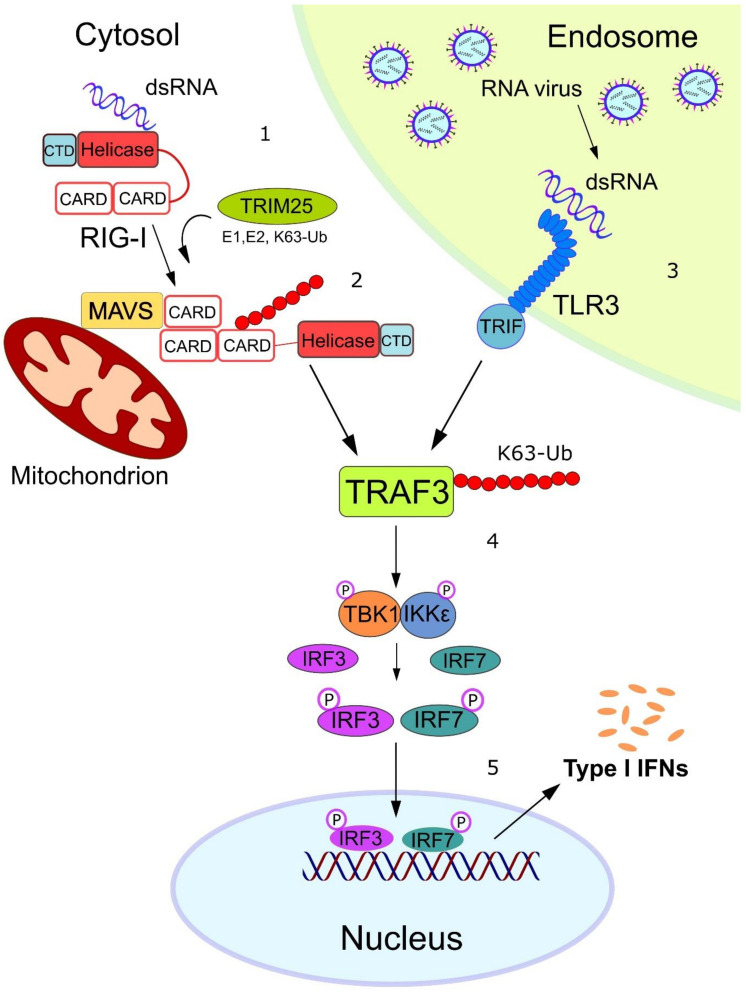
An RNA virus activates the type I Interferon response. **1**. The cytosolic RNA sensor RIG-I undergoes conformational changes after binding to short double-stranded (ds)RNA through its helicase domain and C-terminal domain (CTD). This event permits the N-terminal caspase activation and recruitment domain (CARD) of RIG-I to be exposed for TRIM25-mediated K63-linked ubiquitylation. **2**. Ubiquitylated RIG-I can now bind to the CARD domain of mitochondrial antiviral signaling (MAVS) protein, which in turn leads to the recruitment of the E3 ligase TNF receptor-associated factor (TRAF)3 and its subsequent K63 ubiquitylation. **3**. In parallel, long dsRNA can be sensed by the Toll-like receptor (TLR)3 receptor within the endosome, leading to the TRIF-mediated recruitment of the E3 ligase TRAF3, which in turn K63-autoubiquitylates itself. **4**. K63-ubiquitylated TRAF3 acts as a scaffold protein for the recruitment and activation of the kinase complex TBK1/IKKε. **5**. Activated TBK1/IKKε phosphorylates the transcription factors IRF3/IRF7, which translocate into the nucleus where they induce the expression of type-I interferon genes.

**Figure 3 biomolecules-10-00951-f003:**
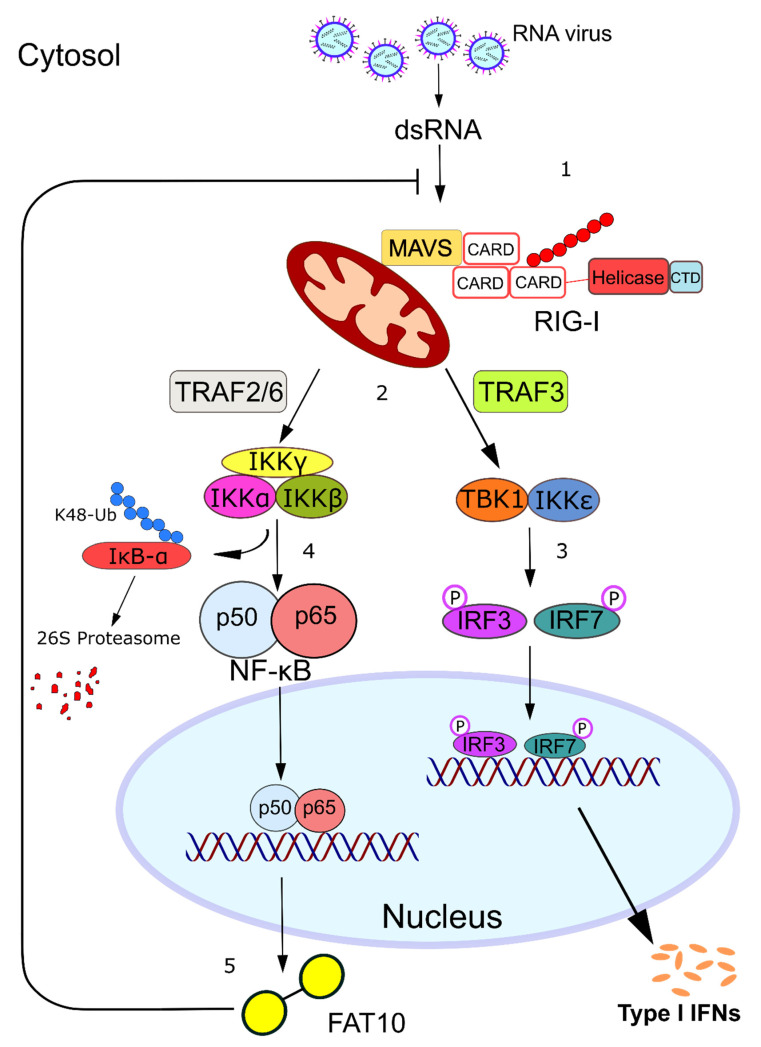
FAT10 is expressed in response to RNA virus infection and acts as a negative feedback regulator of RIG-I activation. **1**. dsRNA is sensed by cytosolic RIG-I, which is activated and interacts with MAVS. **2**. The generated MAVS filaments act as a platform for the recruitment of the E3 ligases TRAF3 and TRAF2/TRAF6. **3**. TRAF3 is responsible for the activation of the TBK1/IKKε complex, which leads to the expression of type-I interferons. **4**. The recruitment of TRAF2/TRAF6 activates the IKKα/IKKβ/IKKγ complex, which in turn triggers the canonical NF-kB pathway. **5**. *Fat10* is one of the genes that are induced upon NF-kB stimulation, and its expression in turn inhibits RIG-I activation, facilitating viral replication and acting as a negative-feedback modulator of the anti-viral response.

**Figure 4 biomolecules-10-00951-f004:**
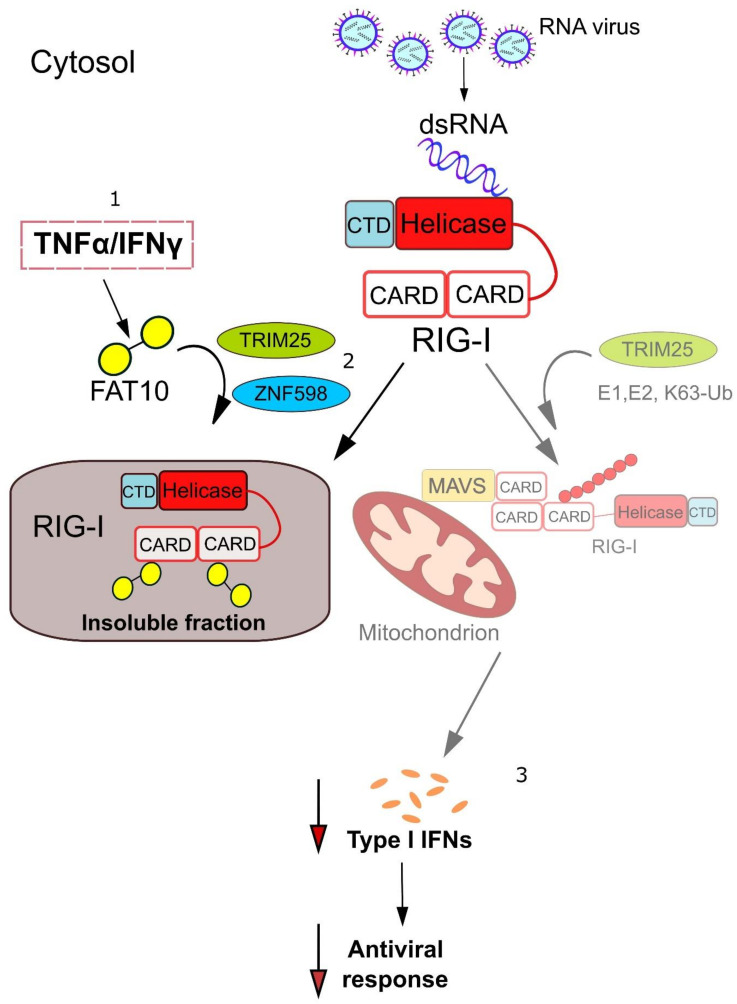
FAT10 inhibits RIG-I activation and impairs the type-I interferon signaling cascade. **1**. The ubiquitin-like modifier FAT10 is expressed upon stimulation with the pro-inflammatory cytokines TNFα/IFN-γ and interacts with the cytosolic RNA sensor RIG-I. **2**. The E3 ligase TRIM25 stabilizes FAT10 bound to RIG-I, which results in its sequestration into a not yet defined insoluble compartment. The E3 ligase ZNF598 promotes the interaction between FAT10 and RIG-I, which in turn reduces the K63-linked ubiquitylation of RIG-I. **3**. This inflammation-mediated inhibition of RIG-I activation impairs the type-I interferon signaling cascade, thus down-modulating the anti-viral response.

**Figure 5 biomolecules-10-00951-f005:**
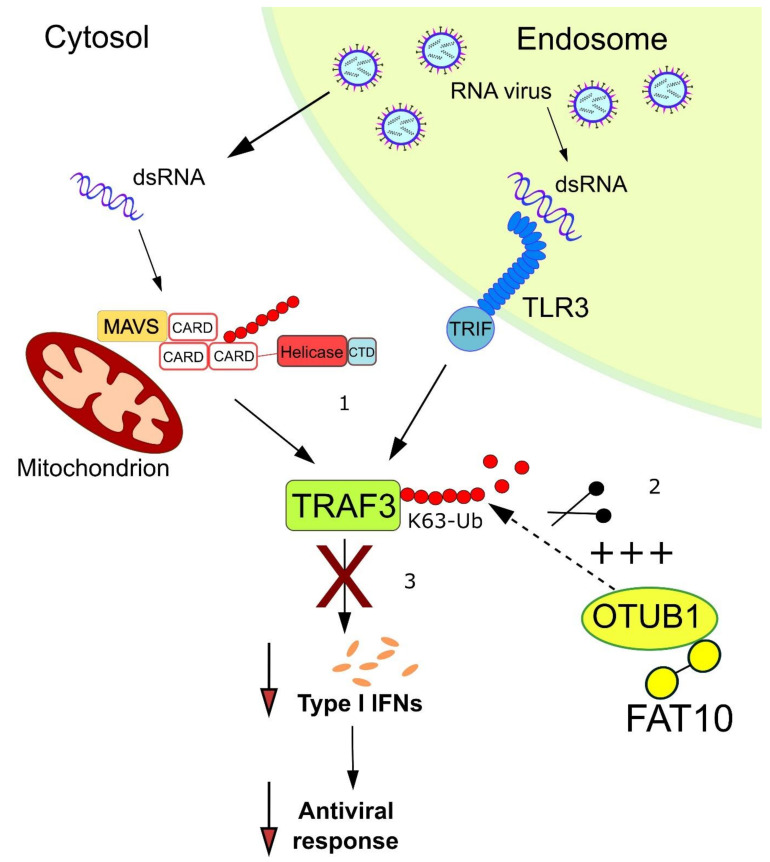
FAT10 stimulates the activity of OTUB1 in deubiquitylating TRAF3. **1.** dsRNA is sensed by the cytosolic RNA sensor RIG-I and by the endosomal receptor TLR3, resulting in the activation of TRAF3 by K63-linked autoubiquitylation. **2**. FAT10 is expressed upon TNFα/IFN-γ stimulation and interacts with the deubiquitylating enzyme OTUB1. This non-covalent interaction enhances the activity of OTUB1, which becomes more efficient in removing K63-linked ubiquitin from TRAF3. **3**. The deubiquitylation of TRAF3 counteracts the optimal continuation of the type-I interferon response, possibly resulting in an attenuated anti-viral response.

**Figure 6 biomolecules-10-00951-f006:**
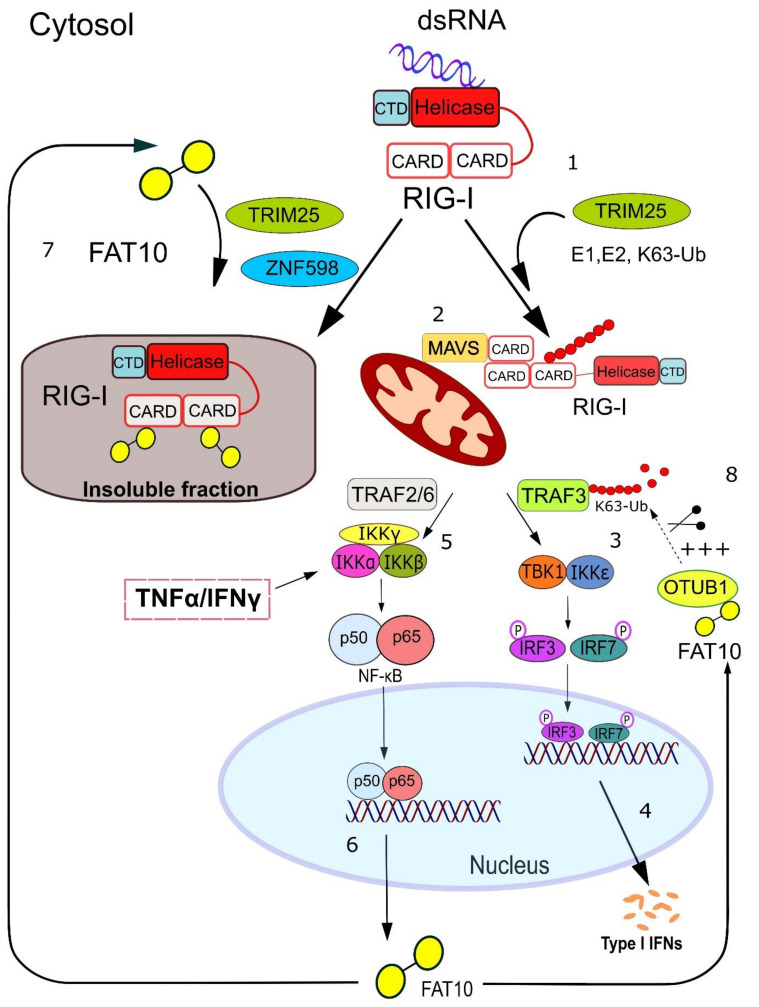
Proposed model of FAT10-mediated regulation of type-I interferon response. **1**. RIG-I undergoes activating conformational changes and ubiquitylation upon binding to viral dsRNA. **2.** Ubiquitylated RIG-I localizes to the mitochondria where it interacts with MAVS. **3–4**. This event recruits and activates TRAF3, which can act as a scaffold protein for the kinase complex TBK1/IKKε. In turn, TBK1/IKKε phosphorylates the transcription factors IRF3/IRF7 which translocate into the nucleus where they induce the expression of type-I interferon genes. **5–6.** In parallel, the activation of RIG-I leads to the mitochondrial recruitment of the TRAF2/TRAF6 ligases, which stimulates the activation of the IKK kinase complex, followed by the degradative ubiquitylation of the NF-kB repressor IκB-α. The consequent nuclear translocation of the NF-kB p50/p65 heterodimer stimulates the transcription of NF-kB-inducible genes, including the *Fat10* gene. Notably, the activation of the IKK kinase complex and the expression of FAT10 can also be induced by the pro-inflammatory cytokines TNFα/IFN-γ. **7.** Here, FAT10 is stabilized on RIG-I by the E3 ligases TRIM25 and ZNF598, and this non-covalent interaction results in the sequestration of RIG-I into an insoluble fraction and/or reduces RIG-I poly-ubiquitylation. **8.** Alternatively, FAT10 positively interacts with the deubiquitylase OTUB1 and enhances its catalytic removal of ubiquitin chains from TRAF3, which results in its deactivation as an interferon signaling molecule. These two FAT10 functions represent the likely mechanism by which the inflammation-induced ubiquitin-like modifier FAT10 impairs type-I interferon synthesis thus modulating the anti-viral response.

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
