# Peer review of "Regulation of Interferon Induction by the Ubiquitin-Like Modifier FAT10"

_biomolecules, 2020, doi:10.3390/biom10060951_

Round 1

Reviewer 1 Report

Regulation of interferon induction by the ubiquitin- like modifier FAT10

Line 13 specify TNFα and not just TNF

Line 15 IFNφ

Line 18 "After introducing FAT10",

Line 24 "… modifier",

Line 31 TNFa

Line 39 Instead of REFs 9 and 10, cite J.R. Gruen et al. in Genomics, 36 (1996), pp. 70-85 and M.M. Mah et al. in Mol. Immunol., 108 (2019), pp. 111-120.

Line 84 "mature dendritic cells (10,29)" and macrophages (Cite Michal Kandel-Kfir et al. in Mol Immunol. 2020 Jan;117:101-109.

Line 85 TNFa

Line 87 TNFa

Line 94 change secretion to production

Line 114 omit the space after "of"

Line 125 omit the space after "displayed"

Line 188 change "instruction" to transduction

Line 193 TNFa

Line 201 TNFa

Line 211 TNFa

Line 213 the term "a not closer defined" is not clear, perhaps change to a not yet defined

Line 219 add a , after "TRIM25"

Line 220 add a , after "activation"

Lines 227-229 rephrase the whole paragraph including the unclear term "a not closer defined". Also, consider referring it to figure 4.

Line 246 change to upon ZNF598 overexpression

Line 251 TNFa

Line 257 conditions,

Line 286 and line 331 TNFa

Author Response

Please see attached WORD file with our point-to-point reply to Reviewer 1.

Reviewer 2 Report

The review is well written with nice schematics to highlight pathways. The authors have also cite relevant literature.

I have few minor comments to be addressed by authors:

1. There is strong in vitro evidence for FAT10 role in regulation of type I and type II IFN responses, however a clear antiviral role of FAT10 in vivo is still lacking. Could authors further elaborate and discuss on this topic, role of FAT10 axis in context of viral pathogenesis especially in context to influenza studies (Zhang et al ref 31). In addition to LCMV and IAV has FAT10 effects studied in context of other viruses?

2. In Figure 2, introduce TRAF3 in main text as well or say further discussed in section 6.

3. The complete line to show negative feedback FAT10 is missing in Fig 3.

4. There are typos that should be fixed- abstract line 15- IFN and  line 234 “ A prior"

Author Response

Please find our point-to-point reply to comments by Reviewer 2 in the attached WORD file.
